# Institutions matter: The role of institutions in the relationship between decision-making power and contraceptive autonomy

Laura Rossouw *, Adeola Oyenubi

School of Economics and Finance, University of the Witwatersrand, Johannesburg, South Africa

* laura.rossouw@wits.ac.za

## Abstract

Several academic papers have shown that the distribution of household decision-making power has a positive influence on women's contraceptive autonomy. This paper considers the role of the social or institutional environment (as captured by place of residence) in ameliorating or contributing to this relationship. Our study focuses on the context of Nigeria, a country with diverse gender norms and religious practices, often determined by geographic location. For example, Western common law is more prevalent in the southern Nigeria, while Sharia law is largely practiced in the North of the country. The analysis uses the Performance and Monitoring for Action (PMA) dataset (2019–2020), and analysis is done using logistic regressions. We find evidence that the relationship between household decision-making power and contraceptive autonomy is mediated by institutions (city). More specifically, we show that in Lagos–a state characterised by the increasing empowerment of women–women have more contraceptive autonomy as they gain decision-making power in the household. The opposite is true in Kano—household decision-making power has no bearing on contraceptive autonomy.

## Introduction

The fertility rate in Nigeria has been steadily declining, from an average of 6.4 births per woman in 1960 to 5.3 in 2019. Despite the incremental decline, this fertility rate is almost double the average in other lower-middle income countries [1]. However, with a population of over 206 million [2], Nigeria is a large and diverse country with 36 states, several ethnic groups and different socio-cultural identities [3], and average reproductive trends may misconstrue state-level tendencies. Most notably, Nigerian fertility rates are reportedly higher in the northern states than the southern states [4]. A key predictor of fertility includes access to modern contraception [5], where modern contraceptive methods include the pill, IUD, injectables, implants, condoms, emergency contraception and male and female sterilization. In 2013, access to contraception by women in Nigeria was still heavily skewed to the southern states at 44 percent, compared to 12 percent in the northern states [6].

**Data Availability Statement:** The authors use the Performance Monitoring for Action (PMA) data set. Given that this is third party data, the authors are

not allowed to share the data set. However, it can be easily downloaded from the PMA website, by registering and providing a brief description of the intended use for the data. The exact location for registering is here: https://www.pmadata.org/data/request-access-datasets. Please note that the current analysis can be completed replicated using this data set and following the steps outlined in the methodology. The authors had no special privileges pertaining to this data set.

**Funding:** The authors received no specific funding for this work.

**Competing interests:** The authors have declared that no competing interests exist.

Access to modern contraceptive methods is a function of several structural, cultural, and socio-economic factors, including women's empowerment and household bargaining power [7–11]. Access to modern contraception not only encompasses the physical availability of these methods, but also their acceptability and affordability. Contraceptive knowledge, access and use are shaped by personal factors (like autonomy, attitudes and feelings about contraception), community influences [12], and the surrounding socioeconomic and cultural environment (i.e. informal institutions) [13]. The role of the cultural environment and informal institutions in shaping women's empowerment and household bargaining power, and therefore, access to modern contraception is an area that has not been evaluated extensively.

In this article, we investigate the relationship between informal institutions in Nigeria (captured by the respondent's city and religion) and contraceptive autonomy, as well as the relationship between household decision-making power and contraceptive autonomy. Finally we explore how institutions and household decision-making power interact to affect contraceptive autonomy. Contraceptive autonomy refers to the ability of women to use their contraception of choice, but also includes the choice to opt out of using any contraception.

The research question will be evaluated within an institutional theory conceptual framework, which posits that institutions shape individuals' behaviours and interactions within a society. According to North, institutions are 'humanly devised constraints that structure human interactions. They are made up of formal constraints (rules, laws, constitutions), informal constraints (norms of behaviour, convention, and self-imposed codes of conduct), and their enforcement characteristics' [14].

In our analysis, institutions are defined as social norms and culture as determined by respondents' geo-location. In Nigeria, women's geographical location is likely to determine their ethnic group, lifestyle and often religion [3]. We propose that institutions like beliefs and community norms play an important role in shaping attitudes and practices related to health and contraception choice, while simultaneously shaping household decision-making power and internal family dynamics.

Institutions may influence household decision-making power and health choices through a myriad of pathways. They may prescribe specific gender role for men and women, which could affect decision-making and the distribution of authority [15]. Institutions often offer moral and ethical guidance, shaping perceptions about right and wrong which may shape household decision-making within a household [16]. Institutions may also offer household access to communities and social networks, where peer effects in turn might affect decisions around education, health and social activities [17]. Local customs as institutions also often have a direct influence on contraceptive choice and autonomy, discouraging the use of contraception or limit the choice of acceptable contraception types [15,18,19].

Our analysis is restricted to two Nigerian cities, namely Kano and Lagos. Kano is situated in northern Nigeria where Sharia Law is largely practiced. Lagos is situated in southern Nigeria where western common law is more prevalent [20–22]. The Hausa/Fulani ethnic groups often reside in the North of the country, while the Igbo and Yoruba tribes reside in the South [23]. In their national and subnational evaluation of female empowerment in Nigeria, Rettig *et al.* find that while female empowerment increased overall in Nigeria since the 1990s, this is largely driven by improvements in the South, with the northern Nigerian states remaining relatively unchanged [6].

Differences in institutions between northern and southern Nigeria can arguably be dated back to pre-colonial days. Prior to the arrival of the British, Arabic traders ventured into the northern Nigeria [24]. Consequently, Hausa people occupying northern Nigeria have a cultural heritage predominated by Arabic influence. The political system, for example, was based on Islamic law [24]. Western influence and religion is more strongly represented in heritage

and the way of life in southern Nigeria. For example, transatlantic slave trade in southern Nigeria pre-dated 1790 [25] while western influence in form of Christian missionaries in southern Nigeria dates back to 1841 [26]. This difference in history affects present reality and in part explains differences in institutional environments in present day Nigeria. The analysis of these two cities situated in different states allows us to investigate the potential role of institutions and social norms in the relationship between household decision-making power and contraceptive autonomy.

## Methods

### Data

For this analysis, we used the Performance Monitoring for Action Nigeria Phase 1 Household and Female Survey Dataset, which focused on contraception, reproductive health, and family planning. The specific Nigerian dataset was chosen as it also explores women's autonomy. The data was sampled using a stratified cluster design with urban-rural strata, based on the National Population Commission's master sampling frame. Data was collected during December 2019 and January 2020. The final sample collected consists of 1122 women in Kano and 1469 women in Lagos [27]. We restricted the sample to women who were married or living in a cohabiting relationship, so as to gauge women's contraceptive autonomy within the context of a sexually active relationship and household. As a result, we have a sample of 788 and 865 in Kano and Lagos respectively.

### Variables

**Dependent variable: Contraceptive autonomy.** Our definition of contraceptive autonomy coincides with the modern perspective on reproductive decision-making. Woldemicael writes that the definition of women's autonomy in reproductive decisions shifted after the 1994 International Conference on Population and Development, from joint or interdependent decision-making with a spouse, to emphasizing the role of woman's individual choice [28]. This follows the argument of Senderowicz (2019), who noted that non-autonomous use of contraception is also a negative outcome when it comes to autonomy and contraceptive use. By exploring contraceptive autonomy, rather than contraceptive use, we are able to capture the nuanced and dynamic nature of contraceptive choice [29].

Contraceptive autonomy was captured using the questions 'Would you say that using contraception is mainly your decision' and 'Would you say that not using contraception is mainly your decision'. For both of these questions, respondents could answer that the decision was either a) mainly theirs, b) the decision of their husband/partner, c) a joint decision or d) somebody else made the decision. Using this information, we created a binary variable equal to one if the respondent is not involved with their contraceptive choices. Either their husband/partner or somebody else made the decision for their contraceptive use or non-use.

**Independent variable of interest: Household decision-making power.** Household decision-making power was established using a set of six questions. Respondents were asked who in the household makes decisions around (1) large household purchases, (2) household purchases for daily needs, (3) medical treatment for the respondent, (4) buying clothes for the respondent, (5) the spending of the respondent's earnings, or (6) how the respondent's partner's earnings are spent. For each of these questions, respondents could answer that they either a) made the decision themselves, b) their husband or partner made the decision, c) they made the decision with their partner jointly, or d) someone else decided.

Using this data, we create a decision-making index that counts the number of instances where the respondent is involved in decision making (either solely or jointly). The index ranges from 1 to 6, with a higher value indicating more decision-making power.

**Covariates.** Our choice of covariates is informed by our institutional theory conceptual framework. We control for several covariates that explain contraceptive autonomy, including marital status ('Currently married', 'Currently living with man') and whether it is polygamous, duration of co-habiting, age, possession of health insurance, ethnicity (Hausa, Igbo, Yoruba, and others), and whether the respondent is the head of household. The respondent's economic and financial position is captured by including a dummy variable equal to one if the respondent has savings, land ownership, earning status (are they renumerated in 'Cash', 'Cash and kind', 'In-kind', 'Not paid' and 'Not working') and the respondents household wealth quintile.

Our institutional variable is state, which is a binary variable equal to one if the respondent resides in Lagos and zero if they reside in Kano. In addition to state, we also include a potential second institutional variable—a categorical religion variable (Christian, Islamic, Other).

Lastly we control for a variable that captures knowledge of contraception created as a summary index ranging from zero to 16, with a value of one every time a respondent answered that they have heard of the following contraceptive methods: 'female sterilization', 'male sterilization', 'implants', 'IUD', 'injectables', 'birth control pills', 'emergency contraception', 'male condoms', 'female condoms', 'diaphragm', 'foam or jelly as a contraceptive method', 'standard days method or cycle beads', 'LAM', 'rhythm', 'withdrawal' and 'others'.

## Data analyses

Our analysis uses logistic regression to estimate the relationship between household decision-making power and the level of autonomy in contraceptive use. We account for the role of institution by interacting state with decision-making power to see if the relationship between the two variables depends on institutional context as measured by state. The premise of this analysis is that the relationship between culture, self, and autonomy suggests that culture (or context) plays a key role in determining the potential for autonomy [30]. Culture influences the definition of self and by doing so, sets the boundaries for what socially acceptable level of self-determination or autonomy [31]. This suggests that differences in cultural contexts between Kano and Lagos may influence the relationship between autonomy and contraceptive use. Stata 16 is used to conduct the analysis.

## Results

The descriptive statistics (Table 1) show substantial differences in the contraceptive autonomy, decision-making power, and economic empowerment between women in Kano and Lagos. While 23.1 percent of women in Kano report lacking contraceptive autonomy, only 8.7 percent report lacking autonomy in Lagos. Further average autonomy index in Lagos is double that of Kano state (4.3 versus 2.3). Kano consists predominantly of an Islamic population (98.9 percent), while Lagos consists of both large proportions of Christian (64.9 percent) and Islamic (33.9 percent) women. Women in Lagos are more likely to own land, work and earn cash, and have health insurance and savings than women in Kano. The descriptive statistics show that, consistent with the literature, Hausa women live largely in Kano, while Igbo and Yoruba women live largely in the South.

In Table 2, we report the results from the logistics regression of our contraceptive autonomy variables onto our set of covariates. In column (1), we regress lack of contraceptive autonomy onto household decision-making power and other covariates, excluding our cultural context and institutional variables (state). State is included in column (2). In column (3), the

**Table 1. Descriptive statistics of dependent and independent variables, disaggregated by city.**

| | | Kano | Lagos | | |
|---|---|---|---|---|---|
| | | Mean | Mean | Difference | P-value |
| No contraceptive autonomy | | 0.23 | 0.09 | 0.14 | 0 |
| Decision-making power index | | 2.261 | 4.284 | -2.023 | 0 |
| Religion | Christian | 0.01 | 0.65 | -0.64 | 0 |
| | Islamic | 0.99 | 0.34 | 0.65 | 0 |
| | Other | 0 | 0.01 | -0.01 | 0 |
| Marital Status | Currently married | 1 | 0.91 | 0.09 | 0 |
| | Currently living with partner | 0 | 0.09 | -0.09 | 0 |
| Polygamous relationship | | 0.46 | 0.12 | 0.34 | 0 |
| Landowner | | 0.12 | 0.26 | -0.13 | 0 |
| Earning status | Cash | 0.53 | 0.83 | -0.31 | 0.00 |
| | Cash and kind | 0.07 | 0.02 | 0.06 | 0.00 |
| | In-kind | 0 | 0 | 0 | 0 |
| | Not paid | 0.01 | 0.03 | -0.01 | 0.04 |
| | Does not work | 0.39 | 0.12 | 0.26 | 0.00 |
| Log of years of cohabiting | | 8.13 | 8 | 0.12 | 0.01 |
| Age in years | | 30.19 | 35.4 | -5.21 | 0 |
| Household head | | 0.010 | 0.03 | -0.03 | 0 |
| Has health insurance | | 0.04 | 0.11 | -0.08 | 0 |
| Has savings | | 0.4 | 0.7 | -0.3 | 0 |
| Wealth status | Quintile 1 (poorest) | 0.21 | 0.13 | 0.07 | 0 |
| | Quintile 2 | 0.2 | 0.21 | -0.01 | 0.55 |
| | Quintile 3 | 0.19 | 0.21 | -0.02 | 0.37 |
| | Quintile 4 | 0.18 | 0.21 | -0.04 | 0.05 |
| | Quintile 5 (wealthiest) | 0.23 | 0.23 | -0.01 | 0.8 |
| Ethnicity | Hausa | 0.84 | 0.02 | 0.82 | 0 |
| | Igbo | 0.01 | 0.19 | -0.18 | 0 |
| | Yoruba | 0 | 0.63 | -0.63 | 0 |
| | Others | 0.04 | 0.17 | -0.13 | 0 |
| Contraceptive knowledge | | 6.61 | 9.79 | -3.18 | 0 |
| Observations | | 788 | 865 | | |

interaction between the decision-making power index and state is included to see how cultural context influences the relationship.

The main results show that those women who score higher on the decision-making power index are less likely to report lacking autonomy when it comes to contraceptive use (Column (1)) a relationship that is significant at the 5% level. Once the state is introduced (Column (2)), the estimate of the decision-making power index is reduced and its relationship with contraceptive autonomy became statistically insignificant. However, state shows a statistically significant relationship with contraceptive autonomy. Specifically, those who live in Lagos are significantly less likely to report lack of contraceptive autonomy (the relationship is significant at 1%). This suggests that institutional environment is important in the relationship between women's empowerment and contraceptive autonomy. Lastly, Column (3) includes the interaction between decision-making index and state, the interaction term is statistically significant (at the 5% level), suggesting that those with higher level of decision-making power living in Lagos are less likely to report lacking contraceptive autonomy compared to women living in Kano.

**Table 2. Logit regressions: Lack of contraceptive autonomy on autonomy in household decision-making power.**

|  |  | (1) | (2) | (3) |
|---|---|---|---|---|
| State | Kano | (Reference) | (Reference) | (Reference) |
|  | Lagos |  | -1.33*** | -0.60 |
|  |  |  | (0.41) | (0.51) |
| Decision-making power index |  | -0.10** | -0.07 | -0.02 |
|  |  | (0.05) | (0.05) | (0.05) |
| Lagos # Decision-making power index |  |  |  | -0.22** |
|  |  |  |  | (0.10) |
| Religion | Christian | (Reference) | (Reference) | (Reference) |
|  | Islam | 0.57** | 0.08 | 0.03 |
|  |  | (0.25) | (0.29) | (0.30) |
|  | Other | 3.38*** | 3.36*** | 3.40*** |
|  |  | (0.67) | (0.66) | (0.67) |
| Marital Status | Currently married | (Reference) | (Reference) | (Reference) |
|  | Currently living with partner | -0.12 | -0.05 | -0.07 |
|  |  | (0.44) | (0.44) | (0.44) |
|  | Landowner | 0.15 | 0.16 | 0.13 |
|  |  | (0.19) | (0.20) | (0.20) |
| Employment | Cash | (Reference) | (Reference) | (Reference) |
|  | Cash and in-kind | 0.41 | 0.32 | 0.24 |
|  |  | (0.31) | (0.32) | (0.32) |
|  | In-kind | 0.77 | 0.66 | 0.59 |
|  |  | (0.90) | (0.90) | (0.89) |
|  | Not paid | 0.05 | 0.11 | -0.04 |
|  |  | (0.52) | (0.52) | (0.52) |
|  | Does not work | -0.26 | -0.27 | -0.26 |
|  |  | (0.18) | (0.18) | (0.18) |
| Polygamous relationship |  | 0.39** | 0.31* | 0.33** |
|  |  | (0.16) | (0.16) | (0.16) |
| Log of years of cohabiting |  | -0.05 | -0.08 | -0.08 |
|  |  | (0.10) | (0.10) | (0.10) |
| Age in years |  | -0.00 | 0.00 | 0.00 |
|  |  | (0.01) | (0.01) | (0.01) |
| Has health insurance |  | 0.61** | 0.69** | 0.74*** |
|  |  | (0.28) | (0.28) | (0.28) |
| Has savings |  | -0.27* | -0.23 | -0.22 |
|  |  | (0.16) | (0.16) | (0.16) |
| Wealth status | Quintile 1 (poorest) | (Reference) | (Reference) | (Reference) |
|  | Quintile 2 | 0.28 | 0.30 | 0.29 |
|  |  | (0.24) | (0.24) | (0.24) |
|  | Quintile 3 | 0.61** | 0.62** | 0.62** |
|  |  | (0.24) | (0.24) | (0.24) |
|  | Quintile 4 | 0.29 | 0.28 | 0.27 |
|  |  | (0.26) | (0.26) | (0.26) |
|  | Quintile 5 (wealthiest) | 0.48* | 0.35 | 0.29 |
|  |  | (0.26) | (0.27) | (0.27) |
| Ethnicity | Hausa | (Reference) | (Reference) | (Reference) |
|  | Igbo | -0.21 | 0.37 | 0.40 |

(*Continued*)

**Table 2.** (Continued)

|  |  | (1) | (2) | (3) |
|---|---|---|---|---|
|  |  | (0.37) | (0.42) | (0.42) |
|  | Yoruba | -0.65*** | 0.29 | 0.33 |
|  |  | (0.24) | (0.38) | (0.37) |
|  | Others | 0.16 | 0.43* | 0.46** |
|  |  | (0.21) | (0.22) | (0.22) |
| Household head |  | 0.60 | 0.67 | 0.65 |
|  |  | (0.48) | (0.49) | (0.50) |
| Contraceptive knowledge |  | -0.01 | -0.00 | -0.01 |
|  |  | (0.02) | (0.02) | (0.02) |
| Constant |  | -1.54** | -0.97 | -1.01 |
|  |  | (0.72) | (0.74) | (0.75) |
| Observations |  | 1,652 | 1,652 | 1,652 |

***p<0.01 **p<0.05 ***p<0.1.

When we observe the relationship between religion and contraceptive autonomy, we initially observe that being Islamic is positively and statistically significantly related to a lack of contraceptive autonomy relative to being Christian. However, the effect disappears when we control for state (columns (2) and (3)).

Fig 1 highlight our main result by showing the marginal effect of the relationship between decision-making and lack of contraceptive autonomy. First, lack of contraceptive autonomy is higher in Kano irrespective of the level of decision-making power. However, at lower levels of the decision-making power index (0 or 1 decision out of 6), there is no significant difference between the partial effect of decision-making on contraceptive autonomy (as indicated by the overlapping confidence intervals). For women whose decision-making index ranges from 2 to 6, higher value of the decision making reduces the probability of reporting lack of contraceptive autonomy. Lastly, while the negative relationship between decision making and lack of contraceptive autonomy is apparent for women in Lagos, the relationship is flat in Kano. This suggests that for women in Kano, autonomy in other spheres of life do not necessarily translate into contraceptive autonomy.

## Discussion

This paper finds evidence of a relationship between state as an institution, household decision-making power and contraceptive autonomy. Household decision-making power may influence contraceptive autonomy through a myriad of avenues. First, decision-making power may reflect financial independence and empowerment, which has been associated with the acceptability of discussing fertility choices and family planning with a partner [32]. Economic decision-making power and education have also been associated with lower levels of unmet contraception needs, likely through better access to health facilities [33]. (Ali & Okud, 2013). Decision-making power improves knowledge of contraception [32], which in turn is related to the decision to use contraception [34,35].

This relationship between household decision-making power and contraceptive autonomy has to some extent been established in the literature. The unique contribution of this paper is the finding that show the overpowering role of institutions in mediating the potential impact that decision-making power can play in improving women's contraceptive autonomy. Although we find a relationship between decision-making power and contraceptive autonomy,

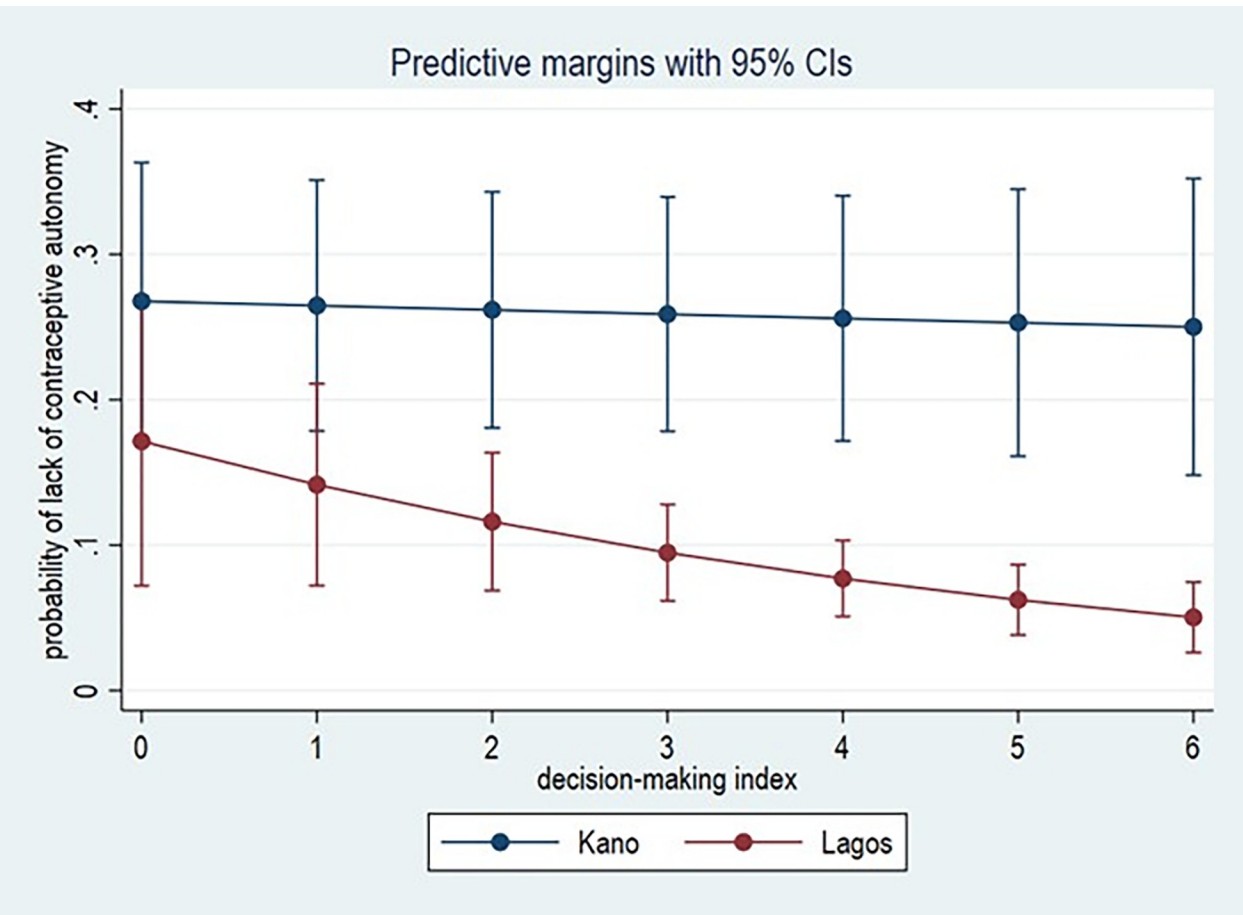

**Fig 1. The marginal effect of the relationship between decision-making power and lack of contraceptive autonomy.**

this relationship is complicated when we control for state–a proxy for institutions. First, we see that at low levels of decision-making power, there is a lack of contraceptive autonomy for women in both Lagos and Kano. However, at higher levels of decision-making power in Lagos–a state characterised by relatively higher female empowerment—women are less likely to report a lack of contraceptive autonomy. The same relationship does not hold in Kano. This suggests that for women in Kano, autonomy in other spheres of life do not necessarily translate into contraceptive autonomy.

Our finding supports the theory that institutions play an important role in establishing the boundaries of women's autonomy, and therefore, their contraceptive autonomy. Although not oft studied, similar findings on the role of institutions have been found in other countries. In India, for instance, one study found that while her education has a positive influence on contraceptive use, the educational level and cultural norms of a woman's community plays a bigger contributing role to her contraception choice [36]. In neighbouring Ghana, Crissman *et al.* (2012) find something dissimilar. The authors find that although individually measured sexual empowerment is a positive predictor of contraceptive use, a regional average of sexual empowerment–a proxy of regional level gender norms around women's sexual empowerment–does not significantly contribute to this relationship [37].

The results also indicate the potential severity of household-level disempowerment of women. Regardless of their community's adoption of specific norms and attitudes towards

contraception, women who lack decision-making power in the household lack choice in various aspects of their daily existence. Autonomy in one area of life often manifests in other areas of life. As pointed out by Crissman *et al.* (2012), the multidimensional nature of empowerment–referring to empowerment across economic, political, sexual, interpersonal and socio-cultural dimensions–mean that empowerment in one sphere does not always equate to empowerment in another sphere [37].

There have been many individual level interventions to improve access to family planning services to reach women alone [38]. For instance, one study in South Africa tested the feasibility of providing women with contraception and counselling at hairdressers, as it is considered a safe, community-based space which is predominantly frequented by women. Offering contraception in this space would allow women to make decisions regarding their sexual and reproductive health independently from their partners [39]. Evidence in Zambia points to a similar strategy in improving contraceptive access. Ashraf *et al.* in a Randomised Controlled Trial tested the difference in contraceptive access when women were offered a voucher to access contraception by themselves, or when access required their partners approval. The authors found that involving spouses in family planning visits resulted in a significant drop in contraceptive access. However, the authors do note that two years after the experiment, there is a significant emotional cost to women who had made their contraception choices in isolation [40].

Our study uses the heterogeneity of norms and religious beliefs across Nigerian states to test the role of this variability on contraceptive autonomy, the findings of which could be relevant to similar contexts in other Sub-Saharan African countries with comparable gender norms, legal systems and religious influences. The generalizability of these findings are limited by several factors. The study focuses on an urban context and the findings might not apply to decision-making power and contraceptive autonomy in more rural contexts, where the availability of healthcare services and contraception may differ. It is also unlikely that these findings are applicable in more egalitarian or secular societies, especially when several policies have been implemented to address gender inequalities.

Future research should be aimed at establishing the robustness of this relationship. This would include applying causal estimation techniques or adopting a more longitudinal approach. A qualitative approach can also be used to get a more in-depth understanding of the interplay between decision-making power, gender norms, religious beliefs and contraceptive autonomy. In addition, by including men in the data collection, and researching their attitudes towards contraceptive use in the same context, one could gain a complementary view of the current results.

## Limitations

Our analysis is not without limitations, importantly we note that the relationships identified are in terms of correlation between the variables of interest and may not have causal interpretation since we are unable to control for unobserved factors. It is also important to note that all concepts are captured using self-reported measures. It is of course possible that institutions such as gender norms or sexual empowerment may affect the accuracy of reporting.

## Conclusion

Institutions, culture, and context can play a key role in empowering women, emboldening their decision-making power, and enhancing their contraceptive autonomy. The paper quantifies the importance of establishing gender equality into discussions around sexual and

reproductive health, even more so in contexts where specific values suppress contraceptive and reproductive autonomy.

## Author Contributions

**Conceptualization:** Adeola Oyenubi.

**Data curation:** Laura Rossouw, Adeola Oyenubi.

**Formal analysis:** Adeola Oyenubi.

**Investigation:** Adeola Oyenubi.

**Methodology:** Adeola Oyenubi.

**Project administration:** Laura Rossouw.

**Software:** Adeola Oyenubi.

**Writing – original draft:** Laura Rossouw, Adeola Oyenubi.

**Writing – review & editing:** Laura Rossouw, Adeola Oyenubi.

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
