## [Decision Letter · Decision Letter 0]

23 Jan 2024

PGPH-D-23-02195

‘Institutions matter: The role of institutions in the relationship between decision-making power and contraceptive autonomy’

Dear Dr. Rossouw,

Thank you for submitting your manuscript to PLOS Global Public Health. After careful consideration, we feel that it has merit but does not fully meet PLOS Global Public Health’s publication criteria as it currently stands. Therefore, we invite you to submit a revised version of the manuscript that addresses the points raised during the review process.

Please address the methodological concerns highlighted by the reviewers. 

We look forward to receiving your revised manuscript.

Kind regards,

Anushka Ataullahjan

Guest Editor

Journal Requirements:

Additional Editor Comments (if provided):

Reviewers' comments:

Reviewer's Responses to Questions

**Comments to the Author**

1. Does this manuscript meet PLOS Global Public Health’s publication criteria? Is the manuscript technically sound, and do the data support the conclusions? The manuscript must describe methodologically and ethically rigorous research with conclusions that are appropriately drawn based on the data presented.

Reviewer #1: Partly

Reviewer #2: Partly

2. Has the statistical analysis been performed appropriately and rigorously?

Reviewer #1: Yes

Reviewer #2: No

3. Have the authors made all data underlying the findings in their manuscript fully available (please refer to the Data Availability Statement at the start of the manuscript PDF file)?

Reviewer #1: Yes

Reviewer #2: Yes

4. Is the manuscript presented in an intelligible fashion and written in standard English?

Reviewer #1: Yes

Reviewer #2: Yes

5. Review Comments to the Author

Reviewer #1: The author should consider revising the title. ‘Institutions matter: The role of institutions in the relationship between decision-making power and contraceptive autonomy’.

Consider expunging 'Institutions matter'. The study outcome does not suggest 'institutions matter.

The objective(s) of this work was/were also not clearly stated,

Reviewer #2: This study applies a novel method to explore the role of institutions in the relationship between women’s decision-making power and contraceptive autonomy. While there are numerous strengths to this endeavor, I advise authors to include justification that the assumptions for causal mediation analysis are met. Alternatively, authors may want to consider other methods, such as structural equation modeling or multilevel models. Please see attached document for further comments.

6. PLOS authors have the option to publish the peer review history of their article (what does this mean?). If published, this will include your full peer review and any attached files.

**Do you want your identity to be public for this peer review?** For information about this choice, including consent withdrawal, please see our Privacy Policy.

Reviewer #1: No

Reviewer #2: No

---

## [Decision Letter · Decision Letter 1]

30 Sep 2024

PGPH-D-23-02195R1

‘Institutions matter: The role of institutions in the relationship between decision-making power and contraceptive autonomy’

Dear Dr. Rossouw,

Thank you for submitting your manuscript to PLOS Global Public Health. After careful consideration, we feel that it has merit but does not fully meet PLOS Global Public Health’s publication criteria as it currently stands. Therefore, we invite you to submit a revised version of the manuscript that addresses the points raised during the review process.

EDITOR: Please provide point-to-point response to the reviewer's comments as a separate document

We look forward to receiving your revised manuscript.

Kind regards,

Tanmay Bagade, Ph.D., MS (O&G), MPH, MHM

Academic Editor

Journal Requirements:

Additional Editor Comments (if provided):

Reviewers' comments:

Reviewer's Responses to Questions

**Comments to the Author**

1. If the authors have adequately addressed your comments raised in a previous round of review and you feel that this manuscript is now acceptable for publication, you may indicate that here to bypass the “Comments to the Author” section, enter your conflict of interest statement in the “Confidential to Editor” section, and submit your "Accept" recommendation.

Reviewer #3: (No Response)

2. Does this manuscript meet PLOS Global Public Health’s publication criteria? Is the manuscript technically sound, and do the data support the conclusions? The manuscript must describe methodologically and ethically rigorous research with conclusions that are appropriately drawn based on the data presented.

Reviewer #3: Partly

3. Has the statistical analysis been performed appropriately and rigorously?

Reviewer #3: Yes

4. Have the authors made all data underlying the findings in their manuscript fully available (please refer to the Data Availability Statement at the start of the manuscript PDF file)?

Reviewer #3: Yes

5. Is the manuscript presented in an intelligible fashion and written in standard English?

Reviewer #3: Yes

6. Review Comments to the Author

Reviewer #3: Thank you for addressing previous recommendations. Please further expand on the discussion and/or conclusion sections to include discussion of the generalizability of the findings and recommendations for further research. Please note, too, several confusing word choices and sentence structure choices, highlighted in the attached.

7. PLOS authors have the option to publish the peer review history of their article (what does this mean?). If published, this will include your full peer review and any attached files.

**Do you want your identity to be public for this peer review?** For information about this choice, including consent withdrawal, please see our Privacy Policy.

Reviewer #3: **Yes: **Laura Hoemeke

---

## [Editor Report · Decision Letter 2]

18 Oct 2024

‘Institutions matter: The role of institutions in the relationship between decision-making power and contraceptive autonomy’

PGPH-D-23-02195R2

Dear Dr Rossouw,

We are pleased to inform you that your manuscript '‘Institutions matter: The role of institutions in the relationship between decision-making power and contraceptive autonomy’' has been provisionally accepted for publication in PLOS Global Public Health.

Best regards,

Tanmay Bagade, Ph.D., MS (O&G), MPH, MHM

Academic Editor